# Vulnerability: An Interpretive Descriptive Study of Personal Support Workers’ Experiences of Working During the COVID-19 Pandemic in Ontario, Canada

**DOI:** 10.3390/healthcare12232474

**Published:** 2024-12-06

**Authors:** Upasana Panda, Monique Lanoix, Rebecca Gewurtz, Sandra Moll, Evelyne Durocher

**Affiliations:** 1Faculty of Health Sciences, School of Rehabilitation Science, McMaster University, Hamilton, ON L8S 1C7, Canada; upasanup@mcmaster.ca (U.P.);; 2Faculty of Philosophy, School of Ethics and Public Affairs, Saint Paul University, Ottawa, ON K1S 5T8, Canada

**Keywords:** healthcare workers, long-term care, occupational health, well-being, qualitative, vulnerability, COVID-19

## Abstract

Background/Objectives: Personal support workers (PSWs) are important healthcare workers providing essential services to thousands of Canadians. PSWs face many challenges that were exacerbated in the context of the COVID-19 pandemic. In this study we explore experiences of PSWs working through the pandemic in Ontario long-term care (LTC) homes by focusing on the vulnerability of such workers. Methods: An interpretive description approach was adopted. Eleven PSWs working in LTC homes in an urban center in Ontario participated in semi-structured interviews between January and May 2022. Thematic analysis of the transcripts was informed by concepts of vulnerability. Results: The results suggest that PSWs experienced inherent, situational, and pathogenic vulnerability. Inherent vulnerability was experienced in relation to risks of contracting the coronavirus working in person with residents, and of experiencing physical and psychological distress in relation to challenging interactions with staff, residents and their superiors. Situational vulnerability was experienced in relation to demanding workloads, which were intensified in the context of the pandemic. Participants expressed feeling undervalued, unappreciated, and disrespected, reflecting experiences of pathogenic vulnerability. The narratives shared by PSWs highlighted how the COVID-19 pandemic added new, and magnified pre-existing, challenges and vulnerability, affecting their health and well-being. Conclusions: Understanding risks faced by PSWs in LTC settings is crucial for developing targeted interventions and policies to support PSWs’ health and well-being, mitigate factors that contribute to their vulnerability and promote the long-term sustainability of this caregiving workforce, ultimately enhancing the quality of care provided to residents in LTC facilities.

## 1. Introduction

Personal support workers (PSWs) have historically operated behind the scenes in the healthcare industry, receiving limited recognition for their crucial contributions [1]. PSWs provide essential care to various populations including individuals living in long-term care (LTC) facilities or in the community, or who are admitted to inpatient care settings [2]. As described by the Ontario Personal Support Workers Association (OPSWA) [3], “Personal Support Workers provide care to any person who requires personal assistance with activities of daily living (ADL). PSWs provide personal care and related paraprofessional services in accordance with an established plan of care. They are typically involved in both personal care tasks and incidental activities of daily living such as housekeeping, meal preparation, socialization, and companionship”.

PSWs’ responsibilities can, include providing personal care, completing household chores, and administering clinical treatments under the direction of a registered healthcare professional; PSWs contribute substantially to the well-being of patients and residents [3,4]. PSWs furthermore play a pivotal role in contributing to the well-being and quality of life of vulnerable populations, including older adults and individuals living with disabilities in LTC settings [1,5]. PSWs, are known by various terms globally [6,7] and titles may include terms such as: certified nursing assistants [8], care aides [9], home health aides [10], and home health workers [11]. In Canada, the nomenclature for PSWs varies across provinces and territories, where such workers may be identified as home health care assistants, home support workers, or home care workers [12,13]. Duties can vary across different healthcare settings as well.

Despite the importance of PSWs’ contributions for individuals and the healthcare system, their work is frequently not well recognized, their positions are often not secure, and they commonly encounter challenging workplace situations [14]. During the COVID-19 pandemic, frontline workers garnered considerable attention for their efforts being labelled as heroes [15], which also brought attention to the role of PSWs in health and social care [16,17] and their risk of catching the virus while attending to residents [18]. Since before the pandemic in 2019 to after the pandemic in 2023 there has been a decrease in PSW workforce in Ontario [17] and across Canada broadly [19]. The loss of PSWs across various settings [20] along with recognition of the precarious nature of their jobs [1,14] and their frequently challenging work conditions [16,21,22] highlights a pressing need for greater acknowledgment, support, and protection for these important workers.

### 1.1. Background and Rationale

In Ontario, PSWs make up 70% of the LTC workforce [23]. Despite their critical role, as in the rest of Canada, PSWs lack a regulatory body in Ontario, which leaves them without standardized guidelines for practice. While efforts have been made to address the lack of a regulatory body for PSWs, such as through the proposed Health and Supportive Care Providers Oversight Authority Act (HSCPOA) introduced by the government of Ontario in 2021 [24], there remains a notable absence of regulatory frameworks akin to those established under the Regulated Health Professions Act, 1991 (RHPA) and its associated legislation, the Medicine Act of 1991 [25]. Unlike these established acts, the HSCPOA Act lacks specific delineation of scope of practice and title protection for PSWs, leaving them without formalized guidelines for their roles and responsibilities within the healthcare system [24].

Many PSWs face poor working conditions, commonly receiving low wages and having limited benefits [26]. Only 41% of PSWs work full-time; part-time and casual PSWs often lack benefits despite working similar schedules as full-time workers [14,27]. Authors of a 2021 study by the Ontario Centres for Learning, Research and Innovation (CLRI) identified that PSWs can struggle with language barriers between them and clients, as well as with situations of client aggression, which can not only hinder meaningful connections with residents but also contribute to burnout [28]. Systemic barriers, such as: insufficient support from superiors and colleagues, refs. [1,29,30] policies and their administration that can contribute to challenging working conditions [12,31,32,33] and overwhelming workloads with staff shortages [16,17,34,35] all potentially contribute to PSWs’ vulnerability, and consequently increase their risk of burnout and emotional strain [36,37]. Such unfavorable conditions in conjunction with job insecurity, feeling undervalued by patients and superiors, being required to meet complex needs of many residents [17,38], and working in situations of workplace violence, which is experienced by PSWs in some settings [12,14], can lead to physical and mental health concerns and increase PSWs’ vulnerability [1,12,26]. These impacts for PSWs ripple outwards, affecting the larger healthcare system through PSW workforce shortages, low PSW job satisfaction, and long-term health issues for PSWs, ultimately leading to higher absenteeism and turnover [39,40]. The COVID-19 pandemic exacerbated such challenges for PSWs in LTCs who, already understaffed, had to follow strict guidelines to protect themselves and residents [41]. While protective measures were put in place and were essential, such measures added to PSWs’ workloads and challenges.

Attention to working conditions during the pandemic heightened concerns about distress among care providers. First defined by Jameton [42], moral distress occurs when individuals believe they should act in a particular way but are constrained from doing so, leading to psychological discomfort [43]. PSWs are particularly vulnerable to moral distress due to the complex demands of their work, where ethical and practical conflicts can prevent them from following through on moral choices [44]. Repeated moral distress can undermine well-being and professional identity, potentially leading to moral injury, emotional strain [45], and burnout [36,37,46], exacerbating workforce instability and amplifying vulnerability.

Previous research on PSWs has primarily centered on exploring descriptive information about PSWs such as their ages, genders, educational backgrounds and number of years of work experience [28,47] or their experiences with challenges at work [12,48,49]. There is a lack of research exploring how these challenges contribute to PSWs’ vulnerability and affect their well-being and the quality of care they provide to residents [33,50]. While some studies about the broader population of LTC workers explore psychological health and safety [46] as well as problem-solving approaches within LTC such as facilitated reflection meetings [51], they often do not focus on PSWs’ unique challenges. The research presented here contributes important knowledge to fill that gap.

### 1.2. Purpose

The purpose of this research is to explore the experiences of PSWs working through the COVID-19 pandemic in Ontario LTC homes, focusing on PSWs’ vulnerability within the workplace.

## 2. Methodology

An interpretive description approach [52] was used to explore the experiences of PSWs. Rooted in clinical practice, interpretive description offers a structured methodology to guide the examination of complex phenomena such as human experiences, social interactions and organizational dynamics while promoting emphasis on practical applications of the findings to real-world settings [53,54]. Aligned with a constructivist paradigm, interpretive description methodology was originally developed for research in the field of nursing, however, its adaptability makes it relevant for exploring interdisciplinary topics [54]. The applied health orientation of interpretive description makes it an ideal method to address the objectives of this inquiry, allowing for a nuanced examination of PSW experiences in long-term care contexts during the pandemic.

The research team consisted of individuals with expertise in the fields of ethics, philosophy, nursing, occupational therapy, mental health and older adult health and social care including LTC.

### 2.1. Research Context 

PSWs working in four LTC homes located in a large urban city in Ontario were invited to participate in the study. Digital posters were emailed to the managers of four publicly-funded LTC homes in a large urban city in Ontario and were circulated to PSWs working in those homes via email. Interested PSWs were asked to contact the research team. The primary author then communicated with potential participants by telephone and shared the study details prior to obtaining informed consent for participation. If the participant gave verbal consent for participation, an interview was scheduled. Ethics approval for the research was received from St Paul’s University (1008-2020-0256) as well as the Hamilton integrated Research Ethics Board (HIREB # 13667).Data generation took place during the fifth wave of the COVID-19 pandemic in Ontario, Canada, between January and May 2022. COVID-19 infection rates were high in this region at that time [55]. Multiple regulations and restrictions were put in place to control the spread of infection as well as to maintain the safety of the public [23]. Such regulations included mandates for: the use of personal protective equipment (PPE) such as N95 or approved non-fitted masks, eye protective gear, visors, gowns, and gloves; adherence to hand hygiene and infection control procedures; vaccination; and outbreak management procedures [56]. LTC homes in which study participants worked were placed in lockdown [57] at various points during data generation in relation to preventing spread from an outbreak in the LTC homes or from the community into the LTC homes. A further regulation intended to curb the spread of the coronavirus between facilities restricted PSWs from working in more than one LTC facility between April 2020 and March 2022 [58]. This restriction contributed to many PSWs working fewer hours and contributed to increased distress for some workers who needed more financial compensation than could be provided through the limited hours [1,16].

### 2.2. Data Generation

Semi-structured interviews were conducted by telephone or Zoom in French or English, according to each participant’s preference. Verbal consent was obtained from participants at the time of the interview. The interview guide questions were created by the research team in collaboration with healthcare partners from the LTC system in the region of the study and were focused on exploring experiences of personal support workers working through the pandemic. Questions asked in the interviews included for example: Can you tell me what you found most difficult working during these past few months of the pandemic? How have you dealt with these challenges? Have you felt vulnerable or unsafe? If so, in which situations? Interviews were audio-recorded and lasted between 20 and 40 min (average 30 min). Interview recordings were transcribed verbatim and French transcripts were translated for analysis.

### 2.3. Analysis

The interview transcripts were analyzed using interpretive description methods, which include taking an inductive approach, practicing reflexivity to reduce bias, and applying existing theories to interpret the data [53]. Open coding and theme development approaches were consistent with interpretive description methodology as described by Thorne [53] and were facilitated by using Quirkos software for data management. All transcripts were initially reviewed twice by two members of the research team and a codebook was developed. The initial codebook was iteratively refined by the same two members of the research team after the independent coding of three transcripts. Codes were discussed and categorized with input from team members, enhancing analysis depth. Regular discussions with the research team, the keeping of an audit trail and the practice of reflexivity during the analysis process promoted rigour in the research.

The analysis was informed by the sensitizing taxonomy of vulnerability as described by MacKenzie, Rogers, and Dodds [59]. These authors categorize vulnerability into three types: inherent, situational, and pathogenic. Mackenzie, Rogers, and Dodds [59] refer to inherent vulnerability as related to universal human dependencies, such as physical and emotional needs, which if unmet render individuals susceptible to harm. Situational vulnerability according to these authors arises from external circumstances, like poverty or homelessness. Pathogenic vulnerability, a subset of situational vulnerability, stems from morally harmful factors such as systemic challenges and attempts to reduce vulnerability that paradoxically increase it [59].

## 3. Results

### 3.1. Description of the Sample

Eleven participants were interviewed for the study. According to their preferences, seven participants were interviewed in English and four in French. Ten participants identified as women and one as a man; all had education in healthcare-related fields. Seven participants had completed some type of formal training or course certificate specific to being a PSW. Seven of the participants worked full-time, three worked part-time (two of whom worked full-time hours despite being hired in part-time roles), and one participant worked casual hours. In order to maintain confidentiality, participants were assigned a participant identification code (P1 to P11). Please refer to Table 1 for detailed participant characteristics.

### 3.2. Results of the Analysis

The experiences of the PSWs interviewed in this study will be described in relation to the different types of vulnerability described by MacKenzie, Rogers, and Dodds [59]—inherent, situational, and pathogenic—as identified through the analysis. We acknowledge that the challenges faced by PSWs and the vulnerabilities we identify are interrelated and overlap; we do not claim that the themes presented below are mutually exclusive. The results of the analysis are presented in these three themes however to facilitate an examination of the experiences described by participants and types of vulnerability we identify in the analysis. The identified themes presented below are:(1)Risks to personal safety contributing to inherent vulnerability;(2)Contextual factors contributing to situational vulnerability;(3)Workplace dynamics contributing to pathogenic vulnerability.


**Risks to Personal Safety Contributing to Inherent Vulnerability**


Inherent vulnerability is defined by MacKenzie, Rogers, and Dodds [59] as vulnerability that arises from one’s humanity and corporeality, and thus having physical, affective and social natures, and needs. Inherent vulnerability, according to Mackenzie and colleagues, is also related to humans’ dependence on others [59]. Inherent vulnerability can manifest as dispositional or occurrent whereby dispositional inherent vulnerability is having qualities that could make one vulnerable but not experiencing that particular vulnerability; occurrent inherent vulnerability is not only having the qualities but also experiencing that particular vulnerability at that moment. PSWs who participated in this study, described scenarios that were challenging in the context of the COVID-19 pandemic, and in which their experiences reflected inherent vulnerability to harm and burnout. These experiences included the risk of contracting COVID as well as risks included in situations where interpersonal interactions could lead to physical or emotional harm.

▪
*Risk of contracting COVID-19*


When asked about their experiences, participants talked about feeling vulnerable due to the risk of contracting the virus and of potentially spreading it to their families and loved ones at home and in their personal lives, highlighting both their own and others’ inherent vulnerability. While everyone was inherently vulnerable and in a dispositional state that they could contract the coronavirus, PSWs were inherently vulnerable in an occurrent state of vulnerability because they were working in direct contact with people broadly but also specifically with people who might be infected. As expressed by P4 “I’m afraid to go [to work] and bring COVID-19 here in the house [to] my family, and [to my newborn grandson]”. Despite taking protective measures, PSWs reported feeling the risk of exposure was high, leading some to isolate from loved ones to reduce transmission. P8 for example shared her fear of infecting her family and the additional measures she took to protect them:
“With the family, yes, there was enormous fear. When you work in an environment where COVID-19 is present…It was not easy. Nobody should speak to me. Nobody should touch me. Nobody should … I have to disinfect myself right away from head to toe. Oh my God I had to wash my hair every day. It was impossible…So, there was really a fear at the family level”.

Participants described having to distance themselves from their family and that this added to feelings of stress and burnout.

Participants reported that during lockdown phases of the pandemic, they had to follow safety protocols, including wearing additional PPE. However, participants reported issues with the quality and availability of PPE, which affected their sense of safety. P6 for example stated “masks, those gowns, I mean, all the PPE [made us feel] hot… so uncomfortable for us. And some of the PPE was very cheap. They really were not meant for the medical field.” A few participants mentioned that there was a lack of PPE supplies and resources, which made them feel less safe and more vulnerable to catching COVID-19 at work. This lack was described by P11:
“I have a friend who was working in another home somewhere, she was concerned that sometimes they didn’t have any gloves. When COVID-19 started they sometimes required them to put on two pairs of gloves [also] there weren’t enough masks. So, they did not have the equipment with them at all times”.

Inadequate protective equipment while working closely with COVID-positive residents, increased PSWs’ inherent vulnerability to catching the coronavirus.

▪
*Interpersonal interactions leading to physical and emotional harm*


Two participants described unpleasant interactions with residents or their families. These included violent or aggressive behaviours that made them feel vulnerable to physical and emotional harm. P11 shared distressing experiences saying: “you take [a resident] to the washroom, and suddenly, she hits out of nowhere… physical violence.” and “you just come to give them lunch, he speaks… in a state of dementia, he yells, yells… spitting saliva and things get on you”. The aggression described was both physical and verbal. Participants attributed such behavior to various factors. P9 expressed compassion saying, “certain people who do not accept their illness can become violent”. Other participants suggested such violence was a reaction to challenging experiences of lockdown circumstances.

Two participants discussed challenges in dealing with residents’ families describing that some families made them feel unappreciated. P11 for example shared:
“there are other families, on the other hand, despite everything that you do, they find that you do nothing and make accusations. … it’s that part which hurts a little, which preoccupies us, … [We] give good service, give care to residents, but sometimes, the families—not all of them, ok, there are some that are very grateful, who appreciate what you do… When you are working with stress and all that [and] the family is behind you, say[ing], “You did that.” “You didn’t do that””.

Feeling judged and underappreciated was reported to affect participants’ morale, increasing their vulnerability to emotional and psychological harm.

In addition to interpersonal stressors, participants reported experiencing physical strain from intense workloads. P8 said, “when you get home at night, you hurt everywhere.” P11 echoed this sharing, “today it’s the feet. Today it’s the back… It is work that requires a lot of physical, mental, and moral effort.” The demanding workload increased PSWs’ susceptibility to physical harm, such as pain and fatigue.

Overall, the COVID-19 pandemic heightened PSWs’ inherent vulnerability. Providing care while navigating difficult interactions and dealing with intense workloads left workers vulnerable to both physical and emotional harm, illustrating the complex challenges they faced during this period.

▪
**Contextual factors contributing to situational vulnerability**


Situational vulnerability, according to MacKenzie, Rogers, and Dodds [59], is vulnerability that arises from external circumstances or environmental factors that may be short-term, enduring or intermittent and that increases an individual’s susceptibility to harm or adverse outcomes. Like inherent vulnerability, situational vulnerability can be dispositional or occurrent, meaning again that an individual may be in a situation in which they could be vulnerable but do not actually experience that vulnerability (dispositional) or they can be in a situation in which they are also experiencing the vulnerability (occurrent) [59]. The risk of contracting the coronavirus reflected inherent vulnerability while the context of the pandemic reflected PSWs’ situational vulnerability. In this study, participants shared examples of how the context of the pandemic increased their workload (e.g., following additional COVID-19 safety protocols, working short-staffed) and how the increase in workload in this situation exacerbated their inherent vulnerability to adverse consequences such as burnout, stress and fatigue.

On the one hand, participants reported appreciating that extra safety measures were put in place and feeling protected by them during the pandemic. P8, for example, described that management was proactive in terms of managing safety at work saying: “Management really kept an eye on [security measures], disinfecting your hands, wearing masks.” P6 described their managers as “wonderful” elaborating “they were trying to do the best.…I see how much managers care…. Our director, how much they do for us. How much they encourage us, for us to be safe.” The additional measures mandated by the facilities appeared to create a sense of support and security for the participants, making them feel safer and less vulnerable.

On the other hand, participants described that the safety protocols increased their workload and stress. P10 discussed the added tasks of donning and doffing PPE and the increased time these required saying: “it was hell…between each resident, we had to change everything, the mask, the gloves, the gown, the visor, so it is longer…you lost at least five minutes taking off the PPE, putting the PPE back on.” Participants also described that in addition to increasing the time required to provide care, wearing the extra equipment was disagreeable. P2 stated “the visor really gets in the way… it was very stressful.” P8 stated being more distressed by the PPE than the risk of contracting COVID saying: “I wasn’t able to breathe… I wasn’t afraid of COVID, but rather of what the mask was doing to me.” PPE was reported to be an added burden, getting in the way and increasing stress.

Participants reported being overwhelmed by their increased workloads and being unable to give sufficient time to residents. P6 explained, “I cannot be a good listener because I know my time is short.” P6 further described trying to be cheerful for the residents despite everything saying, “you still must bring smiles to residents’ rooms”. PSWs described feeling compelled to maintain a positive demeanor for residents, which under the circumstances was emotionally taxing.

Participants further reported workload challenges and inability to adequately provide care to all residents were exacerbated by staff shortages. P10 shared there were 40 residents for five or six PSWs, which they reported to be inadequate. P11 noted, “When there aren’t enough staff… it makes the work more difficult.” All participants, except one, emphasized the need for more PSWs to manage the workload. P8 clearly described the tension between competing demands from residents for the PSWs’ limited time and how this affected their work saying, “you are occupied with one, and on the other side someone else needs you… there is really a time problem… things are not being done well.” Overwhelmed by the demanding workload and pandemic conditions, PSWs expressed experiencing emotional distress and burnout, highlighting their inherent and situational vulnerability. P2 stated that, “we do most of the care, but I feel like we’re the ones getting burnt out…balancing everything and stress levels… staff here, they’re burnt out. They’re done…it was really stressful”.

All of these accounts suggest the context of the pandemic exacerbated tensions between providing quality care and managing workloads, leading to situational vulnerability, distress, and burnout among PSWs.

▪
**Workplace dynamics contributing to pathogenic vulnerability**


Pathogenic vulnerability, as described by MacKenzie, Rogers, and Dodds [59], refers to vulnerability in light of abusive interactions or adverse socio-political factors. Pathogenic vulnerability was reflected in the accounts in descriptions of feeling undervalued and not adequately compensated for their work.

Participants described feeling undervalued and unsupported and that this led to feelings of demoralization. They described that despite being told their contributions were invaluable, their requests for help were often overlooked. P2 shared, “the nurse would always say, ‘I’ll help you,’ but when you needed her help, she was doing whatever she was doing… They’re telling us they care about us, but you don’t feel that—you really don’t.” P2 further described their concerns for patient safety were not acknowledged by management when they were assigned to care for both COVID-19-positive and negative residents, thus putting residents at greater risk. P7 affirmed “management needs to listen more to the people actually working on the floor.” Participants described feeling unheard despite working so closely and having the most direct contact with residents.

Participants highlighted the lack of paid sick leave as a systemic issue, one that increased their pathogenic vulnerability. P11 noted that part-time workers lacked benefits like sick leave: “with us, those who were uncertain casuals… you don’t have sick leave; you don’t have paid leave. You don’t have anything.” Although government support existed during the pandemic, accessing it was reported to be challenging. The lack of financial support if they were not working forced workers to attend work while sick, increasing risks to themselves and others.

These experiences illustrate how feeling unheard and undervalued heightened PSWs’ vulnerability, thus contributing to pathogenic vulnerability. When workers feel unsupported, it undermines their morale, contributing to emotional distress, burnout, and diminished job satisfaction. This sense of moral injury further compromises the quality of care they can provide.

Overall, participants’ reports suggest they experienced an increase in inherent, situational and pathogenic vulnerabilities due to the COVID-19 pandemic. They articulated a pressing need for increased support, spanning from increasing staff numbers and enhanced training to improved avenues for seeking guidance from superiors. These calls for support were aimed to address various challenges, including navigating interpersonal dynamics with residents and colleagues, fostering better communication and respect from management, allocating adequate time and assistance to uphold task quality, achieving workload balance, and fostering a greater sense of appreciation and value for their contributions.

## 4. Discussion

Beneath the surface of philanthropic applause and cheers for healthcare workers [60] frontline PSWs face significant and often overlooked vulnerability and challenges working through the pandemic. In this study we aimed to highlight factors that contributed to and exacerbated PSWs’ vulnerability during the COVID-19 pandemic. The findings of this study suggest PSWs experienced inherent, situational, and pathogenic vulnerability during the COVID-19 pandemic.

PSWs working closely with vulnerable populations in LTC settings placed them at high risk of exposure to COVID-19, thus reflecting what Mackenzie and colleagues [59] term inherent vulnerability. PSWs in this study expressed deep fears of contracting the virus and potentially transmitting it to their families throughout the pandemic, a concern voiced by healthcare workers globally [61]. The emotional toll of navigating these fears, while providing care for residents, was identified in this study to create a constant psychological strain for the participants; the pressure to continue working, despite the risks, intensified vulnerability and lead to heightened anxiety and stress among PSWs, which had negative implications for their health and wellbeing.

Situational vulnerabilities arise from external factors that influence an individual’s experience of stress or risk [59]. Several factors in the pandemic context contributed to PSWs’ situational vulnerabilities, including most notably severe staffing shortages in LTC homes, which predated, but worsened dramatically during, the COVID-19 pandemic [62,63]. As reported in this study and the literature [62,63]. PSWs frequently had to care for a disproportionately high number of residents due to colleagues being ill or required to isolate. This imbalance placed immense strain on PSWs, amplifying their workload and reducing the quality of care they could provide, a finding that is in alignment with research focused on PSWs prior to [38] and during the pandemic [21] as well as research conducted across several countries in Europe [64]. The introduction of new infection control measures, such as enhanced personal protective equipment (PPE) protocols [65,66] were intended to increase everyone’s safety, however, adherence to these measures depended on whether the workers received support from their organizations and superiors and if they had enough time and resources to follow them [67]. The participants in this study reported that additional measures that were put in place significantly increased the time required for their routine tasks, hence making it difficult for them to follow them and adding onto their physical stress. Authors of a study conducted during the pandemic focused on PSWs’ compliance of using PPE and following safety measures in Ontario [67], reported that the increased number of tasks and the time required to do them contributed to the experience of situational vulnerability for workers [59] in the form of heightened physical and emotional exhaustion for PSWs, a common experience among healthcare workers during the pandemic [67]. Similar findings were reported by authors of a study conducted in Netherlands in which it was reported that PSWs working in LTCs faced emotional stress and physical exhaustion due to their work demands during the pandemic [22].

Beyond the inherent and situational vulnerabilities, PSWs also experienced pathogenic vulnerabilities—instances where the systems designed to support individuals paradoxically increased their stress [59]. For instance, as it is shown in the analysis, while policies mandating the use of PPE were intended to protect all individuals including PSWs, however, the increased demands these inherently included in combination with inadequate staffing levels to accommodate these new procedures exacerbated PSWs’ stress. Pathogenic vulnerabilities also manifested in feeling of undervalued and unsupported, all of which contributed to moral injury and burnout among PSWs. While the findings of this study are focused on the pandemic context, these issues were already present. In a qualitative investigation predating the COVID-19 pandemic by a decade, Jakobsen and Sorlie [68] interviewed 23 care workers to determine how they felt during ethically challenging situations they encountered at work. These authors identified that workers reported they felt unheard, when they noticed that their concerns about certain ethically challenging situations were not addressed. The workers in that study reported this lack of action made them feel enraged, powerless and led to demoralization in relation to the work they provide [68]. Similar findings were reported in a study conducted in Sweden where nursing assistants who have similar roles to PSWs also shared they feel abandoned, disrespected and undervalued while working in LTCs during the pandemic [69]. In another study also conducted prior to the pandemic in Canada, Braedley, Owusu, Przednowek, and Armstrong [46] highlight detrimental impacts of feeling undervalued and unsupported on the mental health of LTC workers, including PSWs even before the pandemic situation. These authors suggested that individuals in caregiving roles who perceive themselves as undervalued or unsupported are more susceptible to experiencing moral injury, a profound psychological distress resulting from actions, or the lack thereof, that transgress deeply held moral beliefs or expectations [46]. This moral injury can manifest as feelings of betrayal, guilt, and disillusionment, ultimately leading to emotional exhaustion and burnout [46].

During the COVID-19 pandemic, moral injury was an emerging and troubling phenomenon for PSWs [70], particularly when PSWs were placed in positions that conflicted with their values and sense of duty [71]. The emotional distress experienced by PSWs who participated in this study during the COVID-19 pandemic was multidimensional, stemming from both personal and professional sources. PSWs in this study were caught in a difficult position: while deeply concerned about their own health and the health of their families, they also felt an acute responsibility to care for vulnerable residents, many of whom were at high risk of severe outcomes from the virus. The fear of contracting COVID-19 was a constant source of anxiety for these PSWs, contributing to inherent vulnerability. These vulnerabilities were exacerbated by many PSWs having to manage workplace challenges while feeling undervalued by their employers, residents with whom they worked and sometimes their colleagues. Authors of a study conducted in Ontario LTCs reported that the lack of recognition, particularly in terms of financial compensation or tangible support, only intensified feelings of frustration and emotional fatigue in PSWs [16], which was also true in the case of the participants in this study.

Occupational stress among PSWs during the pandemic was compounded by systemic issues such as understaffing, inadequate resources, and increased workloads [16,28]. Staffing shortages, which were already a critical issue in LTC facilities before the pandemic [72] worsened significantly as many workers fell ill, had to quarantine, or left the workforce due to the dangers of the virus [62]. As a result, PSWs found themselves responsible for more residents, often without sufficient time or support to provide adequate care [62,73] challenges that were echoed by the participants in this study. The extreme time pressure, contributing to PSWs having to rush through their duties, were identified by Hapsari and colleagues [16] to further exacerbate PSWs’ stress levels. The PSWs who participated in a study exploring views of the PSWs on the relationship between quality care in LTC and staffing in the US shared that they used time-saving strategies to be able to complete their tasks and cope with the stress of their demanding workload [74]. For example, the authors reported that PSWs chose the clothing for residents rather than waiting for the resident to decide what they wanted to wear to avoid delays [74]. The participants in this study echoed such sentiments and expressed that in situations where care was rushed or incomplete, they were left feeling guilt and frustration, further fueling emotional distress. Anxiety and stress were not uncommon experiences when looking at workers similar to PSWs across the globe. In a review that included studies conducted across Europe, Lethin and colleagues identified that nursing assistants who have similar roles to PSWs experienced anxiety and heightened levels of stress while working through the COVID-19 pandemic [64]. Many such workers in the studies included in that review attributed this stress to lack of support from their management, working short-staffed and feeling undervalued for their work [64]. As seen with the research presented in Canadian contexts of PSWs experiencing psychological distress and anxiety due to occupational stress while working during the pandemic, similar findings have been reported in studies conducted in US where workers who had comparable duties as PSWs also experienced psychological distress and vulnerabilities while working during the pandemic [75].

All of the challenges reported above can contribute to burnout, which in turn can lead to increased absenteeism, turnover, and intention to leave the profession altogether [76,77,78]. These potential implications can further worsen staffing shortages, creating a vicious cycle in which remaining workers have to shoulder even more responsibilities, accelerating the rate of burnout among the workforce [34,76,77,79,80]. The mental health implications of burnout are well documented and significant, and can have long-term consequences such as depression, anxiety, and other stress-related disorders becoming increasingly common among healthcare workers in Canada [81] and in other countries as well [82].

The systemic vulnerabilities that contributed to PSWs’ moral injury, emotional distress, and burnout during the pandemic were not new but had been longstanding issues within the healthcare system prior to the pandemic, particularly in LTC settings [37]. The pandemic magnified these issues, particularly the frequent under-appreciation and undervaluation reported by PSWs [81,83]. Many PSWs reported feeling like second-class even before the pandemic compared to other healthcare professionals, a perception reinforced by lower pay, fewer benefits, and limited access to mental health support [46]. Through the analysis PSWs had to follow additional regulations that were put in place to curb the spread of COVID-19 infection, such as increased adherence to putting on PPE and taking off PPE for each resident which is also reflected as a challenge by other healthcare workers as well [22,84,85]. These regulations were intended to reduce the spread of the virus, but their impact as described by the participants in this study was that these increased care time, which was not accounted in the schedules of the PSWs therefore leading to time constraints and increased workloads, all of which put the workers at risk of experiencing burnout and stress [86]. It is important that at the time policies are being introduced and implemented, there be consideration of unintended consequences and of how the policies may affect those that are directly implicated by the policies.

### 4.1. Implications and Recommendations

This qualitative study provides unique insights into the experiences and vulnerabilities of PSWs who worked in LTCs during the COVID-19 pandemic, insights that go beyond a focus on demographic data and offer several distinct contributions. First, much of the documented research on healthcare workers during the COVID-19 pandemic was focused on regulated healthcare professions such as nursing, physicians, therapists [87,88]; the work presented here sheds light on the important experiences of PSWs. Secondly, while many healthcare workers faced similar stressors during the pandemic [89,90] this research delves into the challenges and vulnerabilities faced specifically by PSWs, allowing for a nuanced examination of their experiences. By focusing on the perspectives of PSWs directly, this research provides rich insights into the specific difficulties they encountered and how these relate to their vulnerability during the pandemic.

This study further contributes to recommendations for how policies and regulations can be tailored to better support PSWs in their roles. One recommendation identified in other work and echoed by the participants here is focused on implementing policies to regulate workload and staffing ratios to ensure manageable workloads and address severe emotional and psychological tolls that the workers experience [50]. Doing so is anticipated to help reduce risks of burnout and fatigue. Mental health support must also be prioritized including access to counseling services, peer support programs, and other resources designed to help PSWs cope with the emotional strain of their work [9,81]. Developing policies that mandate comprehensive training and ongoing education programs for PSWs has been suggested to help to ensure that PSWs have the necessary tools and skills to provide high quality of care [12,50]. Establishing policies to increase wages and provide benefits to all PSW workers regardless of their work type (full-time/part-time/casual), such as benefits and paid sick leave has been proposed in the literature [16] and by the participants of this study to help alleviate some of the risks that PSWs face due to financial burden and increase job satisfaction and retention. Finally, developing polices to ensure the safety and security of PSWs in the workplace including measures to prevent instances of racism, violence, harassment and ergonomic hazards can help to protect PSWs as they complete their tasks [7] and can help mitigate the inherent and pathogenic vulnerabilities PSWs experience. Additionally, as informed by the results of the analysis above, institutional recognition of the value of PSWs is crucial. This includes not only implementing fair financial compensation but also involving PSWs in decision-making processes related to policies that directly affect their work. Doing so can help to reduce feelings of powerlessness and undervaluation that were reported by participants in this study to have contributed to moral injury during the pandemic.

### 4.2. Limitations and Future Work

A limitation of this study is the transferability of findings to other contexts in which PSWs work. The unregulated nature of PSWs and the ambiguity of their roles creates differences in experiences across settings like hospitals, community care, and home care. Although rich data were gathered in this research, the study was limited to four Ontario public LTC homes. Additionally, given the intense pressures experienced by all healthcare providers during the pandemic, only 11 participants were successfully recruited. The small focus on four homes in one region and the smaller sample size are further potential limitations to the transferability of the findings. Future research could include PSWs from other regions in Canada and globally. Another limitation is that only full-time and part-time PSWs were included, excluding on-call workers who often work full-time hours. In future work, understanding the intersecting identities of PSWs could highlight inequities and inform more equitable solutions.

This qualitative study also represents a specific moment during the pandemic, limiting generalizability to non-pandemic contexts. Further research can focus on comparing the experiences of PSWs across different countries and healthcare systems and in different contexts (not during a pandemic or emergency situation) to identify global trends and best practices for supporting PSWs. Next steps in this area of work can also focus on including more collaboration between researchers, policymakers, healthcare organizations, and PSWs themselves to ensure that research priorities are aligned with the needs and priorities of PSWs.

## 5. Conclusions

In conclusion, this study highlights PSWs’ experiences and their perceptions of safety during the COVID-19 pandemic. Factors contributing to their vulnerability included the risk of infection, emotional strain from challenging interactions with residents and staff, and increased workloads due to additional safety mandates. The tension between delivering quality care and facing systemic barriers, such as staff shortages and lack of management support, further heightened their vulnerability to stress, burnout, and frustration which negatively affected their health and well-being. Disrespect and feeling undervalued also demoralized PSWs, yet they continued to provide essential care. The pandemic exposed the significant challenges faced by PSWs highlighting the need for ongoing support beyond crisis situations to support their health and well-being. By amplifying their voices, this study emphasizes the importance of addressing systemic barriers and fostering a supportive workplace to improve the health, well-being and effectiveness of these essential workers. Moving forward, integrating PSWs into decision-making processes and prioritizing resources—such as healthcare benefits, mental health support, training, and adequate personal protective equipment—can create healthier work environments that promote high-quality care. Policy makers must recognize and address the specific needs of PSWs to build a resilient and sustainable healthcare workforce.

## Figures and Tables

**Table 1 healthcare-12-02474-t001:** Characteristics of PSW participants.

Participant ID	Number of Years Worked as PSW in Canada	Part Time/Full Time/Casual	Gender	Age	Education and Relevant Experience
P1	25	Part-time *	Woman	Unknown	Completed PSW level 4 certificates in Canada
P2	15	Full-time	Man	52	Completed PSW course and RPN ** course in Canada, but is not licensed to work as a nurse
P3	22	Full-time	Woman	44	Completed a bachelor’s degree and PSW certificate in Canada
P4	17+	Full-time	Woman	56	Completed a graduate degree internationally and PSW course in Canada
P5	16+	Full-time	Woman	48	Worked as a social worker in Canada, and completed PSW course in Canada
P6	3+	Part-time *	Woman	53	Worked as a nurse assistant internationally and completed PSW course in Canada
P7	9	Casual	Woman	30	Completed university degree in public relations and program coordination and PSW course in Canada
P8	7	Part-time	Woman	45	Completed a diploma program (unspecified) in a Canadian college
P9	12	Full-time	Woman	41	Completed a diploma program (unspecified) in a Canadian college
P10	19+	Full-time	Woman	55	Completed a diploma program (unspecified) in a Canadian college
P11	2+	Full-time	Woman	38	Completed a diploma program (unspecified) in a Canadian college
Average	~13.36 years	7 Full-time, 3 Part-time, 1 Casual	10 women, 1 man	~46.2 years	

* Listed as a part-time worker but worked full-time hours; ** RPN—registered practitioner nurse.

## Data Availability

Restrictions apply to the datasets. The datasets presented in this article are not readily available in order to protect participant privacy and confidentiality. Requests to access the datasets should be directed to the corresponding author: Evelyne Durocher at durochee@mcmaster.ca.

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
