# Peer review of "Vulnerability: An Interpretive Descriptive Study of Personal Support Workers’ Experiences of Working During the COVID-19 Pandemic in Ontario, Canada"

_healthcare, 2024, doi:10.3390/healthcare12232474_

Round 1

Reviewer 1 Report

Comments and Suggestions for Authors

I would like to thank the Editor for the invitation to review this manuscript, which I believe will be of great interest to the readers of Healthcare. While the manuscript is of adequate quality, I believe several revisions are necessary to enhance its overall quality:

  1. The title should also reflect the study design.
  2. Abstract: The abstract is well-structured, but it is overly long and exceeds the word limit based on the journal's guidelines.
  3. Keywords: Please clarify why an asterisk is included next to "Personal Support Worker*."
  4. It would be helpful to provide a more detailed description of the role of "Personal Support Workers" in the first part of the introduction, and to remove this section from the rationale.
  5. After the study rationale, the study’s objectives should be clearly defined.
  6. In the methodology section, provide a more detailed description of the methodology employed (Thorne et al., 2004).
  7. The guiding questions used in the interviews should be clearly outlined in the methods section.
  8. Table 1: Please include the acronyms used in the table legend.
  9. Although the sample size is at the lower end for this type of study, I believe the results are well-presented and clear. However, I recommend including the sample size as a potential limitation.
  10. In the Discussion section, I suggest broadening the discussion by comparing your findings to international contexts. Currently, the discussion only refers to previous studies or theoretical reflections. It would be useful to compare your findings with international settings where healthcare workers perform similar roles.
  11. Implications and Recommendations: Please revise this section to avoid multiple repetitions of "this study."
  12. Ensure the references and in-text citations follow the journal's guidelines.
  13. Format the text according to the journal's guidelines, ensuring all required sections are included, and use the Microsoft Word template provided at https://www.mdpi.com/journal/healthcare/instructions.

Author Response

Comment 1: The title should also reflect the study design 

Response 1: Thank you for this suggestion. We have included the study design in the title as well as the region as per the suggestion of Reviewer 2 .  The title now reads: 

 Vulnerability: An interpretive descriptive study of personal support workers’ experiences of working during COVID-19 pandemic in Ontario, Canada 

Comment 2: Abstract: The abstract is well-structured, but it is overly long and exceeds the word limit based on the journal's guidelines. 

Response 2: We have shortened the abstract to adhere to the journal guidelines and have removed citations as was suggested by other Reviewers.  

Comment 3: Keywords: Please clarify why an asterisk is included next to "Personal Support Worker*." 

Response 3: Thank you for bringing this to our attention. We have rectified this by removing the asterisk.  

Comment 4: It would be helpful to provide a more detailed description of the role of "Personal Support Workers" in the first part of the introduction, and to remove this section from the rationale. 

Response 4: Thank you for this suggestion. We have moved the description of the role of personal support workers from to the introduction and have also added a few sentences to this section to reflect the different terminologies used to describe PSWs in different contexts.   

Comment 5: After the study rationale, the study’s objectives should be clearly defined 

Response 5: We have now included a separate section to state the purpose of this study, this can be found at the end of the background information section on Page 4. 

Comment 6: In the methodology section, provide a more detailed description of the methodology employed (Thorne et al., 2004) 

Response 6: Thank you for this suggestion. We have included additional details to describe the Interpretive description methodology on Page 4 of the manuscript.

Comment 7: The guiding questions used in the interviews should be clearly outlined in the methods section 

Response 7: Thank you for this suggestion. We have included some examples of guiding questions that were included in the interview guide on page 5 of the manuscript. 

Comment 8: Table 1: Please include the acronyms used in the table legend 

Response 8: Thank you for bringing this to our attention, we have included the acronyms when necessary or used full-forms to make it clear. This can be seen on Page 7 of the manuscript. 

Comment 9: Although the sample size is at the lower end for this type of study, I believe the results are well-presented and clear. However, I recommend including the sample size as a potential limitation. 

Response 9: Thank you for this suggestion. We have now included this as a potential limitation under the Limitations section on Page 16 of the manuscript. 

Comment 10: In the Discussion section, I suggest broadening the discussion by comparing your findings to international contexts. Currently, the discussion only refers to previous studies or theoretical reflections. It would be useful to compare your findings with international settings where healthcare workers perform similar roles. 

Response 10: Thank you.  We have included information about the experiences of workers similar to PSWs in international contexts to better contextualize the findings of this study. Please find these additions in the manuscript on Pages 12-14. 

Comment 11: Implications and Recommendations: Please revise this section to avoid multiple repetitions of "this study." 

Response 11: Thank you for bringing this to our attention. The wording has been adjusted. 

Comment 12: Ensure the references and in-text citations follow the journal's guidelines. 

Response 12: In alignment with the journal guideline we have ensured that the referencing and citation throughout the manuscript are in the ACS style.  

Comment 13: Format the text according to the journal's guidelines, ensuring all required sections are included, and use the Microsoft Word template provided at https://www.mdpi.com/journal/healthcare/instructions.- 

Response 13: In alignment with the journal guideline we have included sections that are pertinent to the presentation of our study.  

Reviewer 2 Report

Comments and Suggestions for Authors

Thank you for inviting me to review this manuscript. This qualitative study investigates the vulnerability of personal support workers (PSWs). This well-written paper offers valuable insights into the vulnerabilities faced by PSWs. This paper also aligns well with the scope of healthcare. My comments are as follows:

Title

It's a good title, but I recommend adding the region/country where the study was conducted.

Abstract

1. References should be avoided in the abstract.

2. The abstract is somewhat lengthy and could be condensed. I recommend reducing it to 250 words or fewer, in line with the journal’s guidelines.

3. For the keywords, use terms that are not repeated in the title or abstract. Additionally, the meaning of the asterisk(*) is unclear, so I suggest removing them.

Introduction

1. Throughout the manuscript, please adopt MDPI’s reference style, which uses numbered citations.

2. While the introduction offers a clear overview of PSWs' working conditions and identifies the research gap, it appears to be overly focused on the Canadian context. It could be enhanced by incorporating the situations of PSWs—or similar occupations—within a global context.

Methodology

1. The description of the sampling procedure is insufficient. Please expand on how participants were selected.

2. As this journal follows a single-anonymized peer-review process, please provide the ethical approval number and the names of the approving institutions in the next round.

Results

This section is well-written.

Discussion

Although the discussion is thorough, it appears to be somewhat lengthy and contains overlapping content.

1. For instance, the paragraphs on moral injury, emotional distress, and occupational stress partially overlap with those discussing each vulnerability (paragraphs 2, 3, and 4). It might be more concise to integrate these details within the relevant vulnerability sections.

2. Additionally, general definitions of concepts like moral injury and burnout seem unnecessary in the discussion. For example, most of the sentences in the burnout section could be omitted, as burnout is already a well-recognized risk factor for turnover, mental health issues, etc.

3. The tenth paragraph, starting with “To address the severe emotional~”, would be more appropriate in the "Implications and Recommendations" section.

Conclusion

The study’s findings and implications are well-summarized in this section.

Congratulations on your work!

Author Response

Comment 1: Title-  It's a good title, but I recommend adding the region/country where the study was conducted. 

Response 1: Thank you for this suggestion. We have included both the region as you suggested as well as a description of the study design as suggested by reviewer 1.  The title now reads: 

 Vulnerability: An interpretive descriptive study of personal support workers’ experiences of working during COVID-19 pandemic in Ontario, Canada 

Comment 2: Abstract- References should be avoided in the abstract. 

Response 2: Thank you for this suggestion. We have removed references from the abstract.  

Comment 3: Abstract- The abstract is somewhat lengthy and could be condensed. I recommend reducing it to 250 words or fewer, in line with the journal’s guidelines 

Response 3: We have shortened the abstract to adhere to the journal guidelines.  

Comment 4: Abstract- For the keywords, use terms that are not repeated in the title or abstract. Additionally, the meaning of the asterisk(*) is unclear, so I suggest removing them 

Response 4: Thank you for bringing this to our attention. We have rectified this by removing the asterisk.  We have additionally removed the words that are already included in the title and have added others. 

Comment 5: Introduction- Throughout the manuscript, please adopt MDPI’s reference style, which uses numbered citations. 

Response 5: We have now changed the referencing and citation styles throughout the manuscript to adopt MDPI's reference style.

Comment 6: Introduction- While the introduction offers a clear overview of PSWs' working conditions and identifies the research gap, it appears to be overly focused on the Canadian context. It could be enhanced by incorporating the situations of PSWs—or similar occupations—within a global context 

Response 6: Thank you for this suggestion. We have included some additional details to reflect the different terminologies used to describe PSWs in a global context and different healthcare contexts. This can be found on Page 2 of the manuscript. 

Comment 7: Methodology- The description of the sampling procedure is insufficient. Please expand on how participants were selected 

Response 7: We have included additional details about sampling and participant recruitment under the methods section on Page 5. 

Comment 8: Methodology- As this journal follows a single-anonymized peer-review process, please provide the ethical approval number and the names of the approving institutions in the next round 

Response 8:Thank you for the suggestion. This information has been anonymized for the review process and will be included when the manuscript is ready for publication. 

Comment 9: Results-This section is well-written. 

Response 9: Thank you! 

Comment 10: Discussion-  For instance, the paragraphs on moral injury, emotional distress, and occupational stress partially overlap with those discussing each vulnerability (paragraphs 2, 3, and 4). It might be more concise to integrate these details within the relevant vulnerability sections 

Response 10: Thank you for this comment. We have made edits to streamline the discussion section. Please refer to pages 12-14. 

Comment 11: Discussion- Additionally, general definitions of concepts like moral injury and burnout seem unnecessary in the discussion. For example, most of the sentences in the burnout section could be omitted, as burnout is already a well-recognized risk factor for turnover, mental health issues, etc 

Response 11: We have removed some definitions that were not necessary. Please find the changes throughout the discussion section. 

Comment 12: Discussion- The tenth paragraph, starting with “To address the severe emotional~”, would be more appropriate in the "Implications and Recommendations" section. 

Response 12: Thank you for this suggestion, we have now integrated this paragraph into the Implications and recommendations section.  

Comment 13: Conclusion-The study’s findings and implications are well-summarized in this section. 

Response 13: Thank you for this comment. 

Reviewer 3 Report

Comments and Suggestions for Authors

Dear editor,

I am grateful for the opportunity to evaluate the manuscript healthcare-3277467, entitled “Vulnerability: Personal support workers working during COVID-19 pandemic”. According to the authors, “Background/Objectives: Personal support workers (PSWs) are an important group of healthcare workers who provide essential services to thousands of Canadians. PSWs face many challenges that can increase their vulnerability and that have been exacerbated in the context of the COVID-19 pandemic. In this study we explore experiences of PSWs working through the COVID-19 pandemic in Ontario long-term care (LTC) homes, specifically focusing on factors that contribute to the vulnerability of such workers. Methods: An interpretive description approach (Thorne, 2017) was used to explore experiences of PSWs working through the COVID-19 pandemic and factors that contributed to their vulnerability at work. Eleven PSWs working in LTC homes in a large urban city in Ontario participated in semi-structured interviews conducted by telephone or via Zoom audio between January and May 2022 during the fifth wave of the pandemic in Ontario. Analysis was informed by the Taxonomy of Vulnerability presented by MacKenzie, Rogers and Dodds (2014). Results: The results of the analysis suggest that PSWs experienced inherent, situational, and pathogenic vulnerability. Inherent vulnerability was experienced in relation to PSWs being at risk of contracting the coronavirus working in person with residents during the pandemic, as well as being susceptible to physical and psychological distress related to challenging interpersonal interactions with staff, residents and their superiors. Situational vulnerability was experienced by PSWs in relation to demanding workloads, which were intensified by added tasks in the context of the pandemic. Participants experienced pathogenic vulnerability, expressing feeling undervalued, unappreciated, and disrespected despite their contributions during this challenging time. The narratives shared by PSWs highlighted how the COVID-19 pandemic both added, and magnified pre-existing, challenges and vulnerability, all of which affected their health and well-being. Conclusions: Understanding risks faced by PSWs in LTC settings is crucial for developing targeted interventions and policies to support PSWs' health and well-being, mitigate factors that contribute to their vulnerability and promote the long-term sustainability of this caregiving workforce, ultimately enhancing the quality of care provided to residents in LTC facilities”.

The theme is relevant to public health and in a brief survey, it was not possible to identify similar texts online, which guarantees the potential for citation in an eventual publication. I emphasize that the manuscript has weaknesses that can be worked on to improve the text. For the analysis, I used the COREQ protocol for analyzing and disseminating qualitative data (available on the Equator network website).

- Title: It is short and concise, but could situate the reader in the specific scenario of long-term care homes.

- Abstract: The authors present references in the abstract, which is unusual and in a way unnecessary. Essential information such as the method used was not presented, leaving the reader to intuit that this research has a qualitative proposal, since it deals with interpretative data. 

- Introduction

The term “Personal Support Workers” has different connotations in different parts of the world. It is important for authors to indicate the scope of this professional’s work, whether they are a specialist, an entry-level professional or a support technician. This explanation is essential so that readers from different backgrounds can understand the reality that the authors want to expose.

- Methodology

The authors do not indicate the type of study, although it is clear that a qualitative proposal was used. This weakens the work and erodes the scientific nature of the data, which becomes more informative than academic. Furthermore, I indicate below questions from COREQ that need to be integrated into the text. Please note that many of these questions are already integrated into the text.

1. Which author (authors) conducted the interview or focus group?

2. What were the credentials of the researcher? Example: PhD, medical doctor.

3. What was the occupation of these authors at the time of the study?

4. Was the researcher male or female?

5. What experience or training did the researcher have?

6. Was a relationship established before the study began?

7. What did the participants know about the researcher? For example: personal goals, reasons for developing the research.

8. What characteristics were reported about the interviewer/facilitator? For example, prejudices, assumptions, reasons and interests in the research topic.

9. What methodological orientation was stated to support the study? For example: grounded theory, discourse analysis, ethnography, phenomenology, and content analysis.

10. How were participants selected? For example: convenience, consecutive, sampling, snowball sampling.

11. How were participants approached? For example: in person, by telephone, letter, or email.

12. How many participants were included in the study?

13. How many people refused to participate or dropped out? For what reasons?

14. Where were the data collected? For example: home, clinic, workplace.

15. Was anyone else present besides the participants and researchers?

16. What are the important characteristics of the sample? For example: demographics, date of collection.

17. Did the authors provide questions, instructions, guides? Were they pilot tested?

18. Were repeat interviews conducted? If so, how many? 19. Did the research use audio or visual recording to collect data?

20. Were field notes taken during and/or after the interview or focus group?

21. How long were the interviews or focus groups?

22. Was data saturation discussed?

23. Were transcripts returned to participants for comments and/or correction?

24. How many data coders were there?

25. Did the authors provide a description of the coding tree?

26. Were themes identified in advance or derived from the data?

27. What software, if any, was used to manage the data?

28. Did participants provide feedback on the results?

29. Were quotes from participants presented to illustrate the themes/findings? Was each quote identified? For example, by participant number.

30. Was there consistency between the data presented and the results?

31. Were the main themes clearly presented in the results? 32. Is there a description of the various cases or discussion of secondary themes?

- Results and discussion

The results were well described and presented categories of analysis relevant to the investigation. Discussion is important and uses classic literature on the subject, in addition to recent texts. I suggest incorporating international sources.

Recommendation: Major revisions.

Author Response

Comment 1: Title: It is short and concise, but could situate the reader in the specific scenario of long-term care homes. 

Response 1: Thank you for this suggestion. We have included the region and study design and fear the title has gotten quite lengthy.  We have included long-term care in the keywords.  We hope that you will find these changes to be acceptable.  The title now reads: 

 Vulnerability: An interpretive descriptive study of personal support workers’ experiences of working during COVID-19 pandemic in Ontario, Canada 

Comment 2: Abstract: The authors present references in the abstract, which is unusual and in a way unnecessary. Essential information such as the method used was not presented, leaving the reader to intuit that this research has a qualitative proposal, since it deals with interpretative data.  

Response 2: We have shortened the abstract to adhere to the journal guidelines and have removed citations as was also suggested by other Reviewers. We have clarified the study design and methods.  

Comment 3: The authors do not indicate the type of study , although it is clear that a qualitative proposal was used. This weakens the work and erodes the scientific nature of the data, which becomes more informative than academic. Furthermore, I indicate below questions from COREQ that need to be integrated into the text. Please note that many of these questions are already integrated into the text. 

Response 3: Thank you for this comment.  We have added additional qualifiers about the study design.  We furthermore agree that consideration of tools such as COREQ can greatly increase the rigour of the study.  We have included many details of focus on this tool including details about the expertise of the research team, more details to describe our use of interpretive description for the methodological approach, sample size, and participant recruitment,  the type of data that was collected from each participant, details of the interview process as well as examples of guiding questions, the number of individuals who led the analysis and the use of Quirkos software.  We have used quotes in the Results section to support the themes and sub themes that were identified.  

Comment 4: Results and discussion-The results were well described and presented categories of analysis relevant to the investigation. Discussion is important and uses classic literature on the subject, in addition to recent texts. I suggest incorporating international sources 

Response 4: Thank you.  We have included contextualization of our findings in relation to studies focused on the experiences of workers similar to PSWs in international contexts throughout the discussion section. 

Round 2

Reviewer 1 Report

Comments and Suggestions for Authors

The authors have made appropriate changes to the manuscript, it can be published

Reviewer 2 Report

Comments and Suggestions for Authors

All of my concerns were properly addressed in the revised manuscript. Thank you.

Reviewer 3 Report

Comments and Suggestions for Authors

Dear Editor,

I am immensely grateful for the opportunity to review the manuscript “Vulnerability: An interpretive descriptive study of personal support workers’ experiences of working during the COVID-19 pandemic in Ontario, Canada”.

The text is current and relevant, and the authors clearly indicate the knowledge gap they intend to fill. All of my considerations were met. Among them, I highlight: Changes to the title, changes to the abstract, incorporation of concepts in the introduction, expansion of the method with the steps established by COREQ, expansion of the discussion and reorganization of the references.

The authors presented a rewritten and much strengthened manuscript. Due to the theme and the quality of the material, I indicate my favorable opinion for publication.